# Health Care Expenditure in the European Union Countries: New Insights about the Convergence Process

**DOI:** 10.3390/ijerph19041991

**Published:** 2022-02-10

**Authors:** Claudiu Tiberiu Albulescu

**Affiliations:** Management Department and the Research Centre in Engineering and Management, Politehnica University of Timisoara, 300006 Timisoara, Romania; claudiu.albulescu@upt.ro

**Keywords:** health expenditures, convergence, bounded unit root tests, strict stationarity, EU countries

## Abstract

This paper assesses the convergence process in the health care expenditure for selected European Union (EU) countries over the past 50 years. As a novel contribution, we use bound unit root tests and, for robustness purposes, a series of tests for strict stationarity to provide new insights about the convergence process. We make a comparison between public and private health expenditure per capita and as a percentage of the gross domestic product (GDP), with a focus on six EU countries with different health care systems in place. When we consider the health expenditure per capita, we report mixed findings. We show that the spread from the group average is stationary in the cases of Finland and Portugal when the overall and public expenditure is considered. In terms of private expenditure, the convergence process is noticed only for Austria. For all other countries included in our sample, we document a non-stationary process, indicating a lack of convergence. This result is robust to the different tests we use. However, when we assess the convergence in terms of the health-expenditure-to-GDP ratio, the convergence process is recorded for Austria only. The robustness check we performed using strict stationarity tests partially confirmed the mixed results we obtained. Therefore, our findings highlight the heterogeneity of the EU health care systems and the need for identification of common solutions to the EU health care systems’ problems in order to enhance their convergence processes.

## 1. Introduction

The accelerating ageing process recorded at the European Union (EU) level calls into question the viability of EU countries’ health care systems. Furthermore, the disparities noted in terms of quality of health care services and health-allocated funds put additional pressure on the European health systems, given the ability of EU citizens to choose their service providers. In this context, investigating the convergence process of health care expenditure in the EU is particularly appealing. Indeed, previous studies have failed to provide a clear answer regarding the existence of a convergence process—the movement toward uniformity in terms of health care expenditure and quality of health services. On the one hand, Refs. [1,2] state that there is no convergence regarding health care expenditure in the EU and in OECD countries, respectively. On the other hand, Ref. [3] mentions the existence of a convergence process in terms of EU health care expenditure. 

The issue of health care expenditure convergence in the EU is therefore doubly important. Firstly, the analysis of the convergence of health care expenditure enables assessment of the outcomes of common policies implemented in this area, and particularly of the level of harmonisation in terms of the quality of services provided by different health care systems. Secondly, a thorough analysis of the convergence process contributes to the identification of potential risk triggered by a migration to high-quality health care services within the EU, or by a global pandemic. Consequently, the documentation of a convergence process in the EU health care systems provides noteworthy information about the effectiveness of EU health care policies implemented by the EU member states. In addition, the convergence in health care—a mainstream approach to health care research—means a harmonisation in terms of education, medical research, and capacity to mitigate risks, such as those triggered by the recent sanitary crisis. The absence of convergence simply shows that human health issues are differently addressed in different EU countries, and that the fundamental human rights—namely, the access of people to the health services they need—are not completely achieved. For example, in the context of the COVID-19 pandemic, we have witnessed a different kind of pressure on the national health care systems, although the health outcomes in the EU have improved over the past decade. Different capacities in terms of intensive care units and very different mortality rates raise questions about the intensification of the convergence process within the EU.

Therefore, the purpose of this paper is to provide an updated analysis of the convergence process in health care expenditure in six EU countries—Austria, Finland, Germany, the Netherlands, Portugal, and Spain—covering the period running from 1972 to 2019. Although all of these countries represent “old” EU member states, their accession to the EU happened at different moments in time. For example, in 1972, only Germany and the Netherlands were part of the EU, whereas Portugal and Spain joined the Union in 1986. Finally, Austria and Finland joined the EU in 1995. In this context, we resort to a common method of analysing the convergence process—namely, the investigation of unit root processes in the spread between the health care expenditure recorded in each country and a reliable benchmark (i.e., the group average). This paper is closely related to the paper by [4], and extends the latter work in several ways: First, like [4], we use the bounded unit root tests advanced by [5,6]. Indeed, if the convergence process is assessed in terms of the health-expenditure-to-GDP ratio, the series are bounded, and the use of classic linear, nonlinear, or structural break tests might generate biased results. Likewise, even if bounded series can be considered to be stationary process, the presence of I(1) series can coexist with a bounded process [5,6]. However, unlike [4], we also use a battery of newly advanced strict stationarity tests for robustness purposes [7,8,9], as they are robust to the alternatives of (1) a unit root null hypothesis, or (2) structural changes in the mean and alternatives with unconditional heteroscedasticity. These tests also have good power in detecting changes in higher moments of the unconditional distribution in a time series.

Second, unlike [4], we assess the convergence process both in terms of the health-expenditure-to-GDP ratio and in terms of health expenditure per capita. This way, we are able to better compare our results with the previous findings reported in the literature. Moreover, in order to obtain a global picture of the convergence process, it is recommended to use alternative indicators. Indeed, health care expenditure per capita represents a common indicator to assess the convergence process [1]. Nevertheless, the health care expenditure per capita has a lower volatility, being influenced by the level of economic development [4]. Thus, if the convergence is associated with the effort made by each country to finance the health care sector, then the health-expenditure-to-GDP ratio is recommended.

Third, we analyse the convergence process among the six above-mentioned EU countries for three reasons. Although the health care sectors of these countries rely on the “Bismarckian” tradition, they are very heterogeneous (for more details, please refer to Section 3). Moreover, we use Organisation for Economic Co-operation and Development (OECD) statistics, and the selection of countries is constrained by the availability of data. Our purpose is to study the convergence process over the longest available timespan, and to include in the analysis the beginning of the 1970s, when ample reforms were carried out in the EU countries in terms of health care systems (for additional information regarding the choice of data samples please refer to Section 5.1). Finally, we want to see whether the results reported by [4] still hold considering the recent period, up to the COVID-19 pandemic (the OECD statistics in terms of health expenditures are available for all of the analysed countries up to 2019). As in [4], we compare the convergence process in terms of overall health care expenditure (all financing schemes), governmental expenditure (government/compulsory schemes), and private sector health expenditure (voluntary schemes/household out-of-pocket payments).

The remainder of the paper is organised as follows. Section 2 is dedicated to the literature review. Section 3 provides a short description of the health care systems in the selected countries, underlining the dynamics of health care expenditure. Section 4 presents the methodology for bounded unit root and strict stationarity tests. Section 5 describes the data and the results. The final section concludes and draws policy implications from our findings.

## 2. Literature Review

Using the neoclassical theory of convergence, a number of relatively important studies have analysed the issue of convergence in health care expenditure and provided very heterogeneous results. Given the structure of the analysed country samples, three main categories of studies can be identified in this literature:

Firstly, the most abundant part of the literature deals with OECD countries. For example, Ref. [10] analyses the convergence in health care expenditure per capita of 21 OECD countries from 1975 to 2003. Using Theil’s measure, the study indicates the existence of convergence within the countries, which is explained mainly by the convergence of the health care expenditure as a share of GDP, labour productivity, and employment rate. Ref. [11] identifies both conditional convergence and β-convergence in health expenditure for a panel of 21 OECD countries over the period 1980–2005. σ- and β-convergence in public health care financing (seen both as public financing in % of total health care financing and as a % of GDP) is also identified by [12], using a sample of 23 OECD countries over the period 1970–2005. 

Ref. [13] analyses the stochastic and β-convergence in total health care expenditure as a share of GDP for 11 OECD countries over the period 1960–2006. Using a breakpoint methodology, the results suggest that whereas the stochastic convergence holds for all of the countries, the β-convergence was identified only for four countries before the breakpoints. Similarly, Ref. [14] measures the convergence in per capita health expenditure for 19 OECD countries during 1972–2006. The convergence is identified in 17 out of 19 countries of the panel. Contrary to these results, Ref. [15] describes the non-convergence in the per capita health care expenditure for 19 OECD countries over the period 1970–2005. Non-convergence was also reported by [16], who analysed the “catching up” hypothesis of per capita health care expenditure of 5 OECD countries and the US for the period 1960–2000. The results are sensitive to the use of structural breaks, with the convergence being identified only in this case. Ref. [17] uses a nonlinear asymmetric heterogeneous panel unit root test methodology for a sample of 22 OECD countries between 1980 and 2012, and states that countries as a panel converge to the OECD mean only if the nonlinearity in health care expenditure is explicitly taken into account. Mitigated results were also obtained by the author of [18]—who found that health care expenditure and real per capita GDP for a panel of OECD countries were stationary—and by [19], who found the contrary for a panel of 21 OECD countries. The results of [20] reject the null hypothesis of a unit root for health expenditure and GDP for 20 OECD countries. 

Secondly, studies on the convergence in health care expenditure have been applied to EU countries. Ref. [1] identifies divergence in the health care expenditure within 10 EU countries over the period 1960–1991. The demographic factors as well as the heterogeneity in the level of GDP per capita explain the lack of convergence between the analysed countries. Contrary to these results, Ref. [3] identifies the existence of β-convergence in health care expenditure measured both per capita and as a share of GDP for a panel of 15 EU countries over the period 1980–1995. Ref. [21] found relatively similar results, measuring the σ-, β-, and γ-convergence of the health care expenditure both as a share of GDP and per capita over the period 1992–2004 for a panel of 23 EU countries. By making the distinction between coverage, health care expenditure, and provision, Ref. [22] uses a panel of 19 EU countries over the period 1980–2000, and shows that the convergence is obtained only for private health care financing. Contrary to these findings, Ref. [23] used a nonlinear dynamics methodology, and did not observe any convergence in per capita health care expenditure among the 14 EU countries between 1970 and 2008. 

In addition to the studies dealing with convergence within OECD and EU countries, the literature has developed some analysis of the convergence in health care expenditure measured at the intranational level. Ref. [24] puts forward a relatively slow rate of convergence in per capita health care expenditure across the US states between 1980 and 2004. Ref. [25] found no evidence of overall σ- and β-convergence in health care expenditure among 28 Chinese provinces over the period 1978–2004; nevertheless, their study concludes the existence of 11 convergence clubs, including 2 or 3 members in each club. The same idea of convergence clubs is put forward by [26], testing the degree of convergence in health care expenditure among the US states from 1980 to 2004. While this study found no evidence of overall convergence, it identified the existence of two convergence clubs across US states (including 18 and 32 states, respectively).

Concluding on this very heterogeneous literature, one may say that it is not possible to put forward a unitary vision of the convergence process within developed countries. The aim of our paper is to contribute to this literature using a long data period for a representative sample of EU countries. In addition to these two elements, the originality of our study relies on the methodology used, i.e., the bound unit root tests, whose robustness is checked using strict stationarity tests.

## 3. Health Expenditure, Health Care Systems, and Health Reforms in Selected EU Countries

EU countries are characterised by complex health care systems, mixing public and private financing schemata, where the role of the state is significant. The attention paid to health systems has also increased, as the total health-care-expenditure-to-GDP ratio has doubled over the past 40 years, from 5% on average to 10% at present (please refer to the OECD statistics). This dynamic is explained by a progressive increase in the public spending related to health care systems.

According to OECD definitions, current health expenditure includes personal health care as well as collective services, leaving aside spending on investment. Government spending (based on compulsory health insurance) and private spending (e.g., voluntary health insurance; private corporations) provide a mix of financing arrangements in all of the EU countries. 

Figure 1 shows the structure and the dynamics of health expenditure per capita (Figure 1a–c) and as a percentage of GDP (Figure 1d–f). We can see that the health expenditure per capita increased in all analysed countries, regardless of the expenditure type. The highest level of expenditure per capita was recorded in the Netherlands, whereas the lowest level was recorded in Portugal. An increase in the total-spending-to-GDP ratio for all selected countries might be a sign of increased convergence. However, noteworthy reforms have been implemented in these countries, and the role of public and private financing is changing. For example, the Netherlands experienced ample health care reform in 2006, causing a drop in the private-expenditure-to-GDP ratio from 3% in 2004 to 1.5% in 2007.

At the basis of the health care systems of all of the selected EU economies is the “Bismarckian” tradition. After World War II, following the German example, all of these countries adopted universal health care frameworks, with health insurance supported by social contributions (from employers and employees). At present, Germany and Austria maintain social security systems that provide high-quality health care, but at the same time are very costly. Starting from a similar model, Finland and the Netherlands recently combined a tax-funded municipal model with a national insurance system. The reforms undergone in these countries were not meant to replace the existing framework, but to find parallel solutions for improved quality of health care services. However, Spain and Portugal, the other two countries in our sample, underwent a transition from the Bismarckian model to the Beveridge model—A unitary health care system with universal coverage, where general taxation represents the main source of funding. Below, we present several particularities of each health care system (major reforms are presented in Appendix A), starting with the German case. Information is synthetized from European observatory reports on health care systems and policies in the EU member states.

The German health care system is a universal one, where the traditional model, with a focus on collective goals and efficiency indicators, takes on a new paradigm, wherein the resources are determined by the patient–physician relationship [27]. There is a mandatory insurance scheme where the contributions are paid as payroll taxes (with an employee share approximately equal to the employer share). The self-employed, who are not covered by the statutory health insurance, can request private insurance protection. Germany’s health expenditure as a percentage of GDP is among the highest in Europe.

In Austria, similar to Germany, there is a Regional Health Fund in each state, financed by the federal budget (more than 50%), the local community (more than 25%), and social security institutions [28]. While the establishment of health care funds supposes considerable negotiation with more than 20 social security institutions, physicians, and pharmacy boards, the sanitary supervision is made by the federal authorities. In Austria, approximately 75% of the health care system’s funds come from social contributions, while 25% comes from private sources (i.e., private insurance or direct payments). Austria represents a country with a high level of health expenditure per capita.

In northern countries, the German health care model was implemented in the 1950s and the 1960s, when noteworthy investments were made in building hospitals and in ensuring primary care. The municipal health care system plays an important role in Finland, where municipalities levy taxes and receive transfers from the national system [29]. Partial reimbursement for private health care services also comes from the national health insurance system. Occupational health represents a distinctive funding mechanism. The reforms carried out were meant to find solutions coexisting with the traditional structure of the health care system. For example, the national insurance scheme, which covers all Finnish residents, is today composed of two complementary pools—namely, medical sickness insurance and income insurance. The private insurance covers children in particular. Compared to the other health care systems in our sample, the health-expenditure-to-GDP ratio is reduced in Finland (below 9%). An explanation for this can be found in the low salaries of Finland’s health care professionals. Although this ratio has continuously increased over the past few decades, the share of public expenditure in the overall health expenditure remains approximately the same (80%).

The Netherlands’ health care system presents some similarities with the Finnish system, relying on the German tradition. After World War II, the primary focus of attention was hospital construction [30]. After the oil price crisis in 1973, severe cuts to health care spending were recorded. A small reform took place in 1983, when a coherent provision of primary care services was advanced. However, the system remained practically unchanged until the 2006 reform, when a new health care insurance system emerged, based on risk equalisation. The system is financed through a compulsory scheme for a long-term care, through a basic health insurance framework and through voluntary health insurance. After the 2006 reform, the share of private expenditure dropped considerably in the Netherlands (see Figure 1c).

The Southern European health care systems are characterised by ample reforms. In Portugal, after the 1974 revolution, a considerable restructuration began, with the nationalisation of central hospitals and, subsequently, local hospitals, which had been owned until then by religious charities [31]. During the 1980s and the 1990s, the health reforms were meant to foster the market mechanism, but at the same time, the Portuguese health care system became more public. At present, three overlapping health care insurance frameworks coexist—namely, the universal, the special public, and the private (voluntary) systems, where the private system covers between 10% and 20% of the total population. The predominant universal system is financed through general taxation, and the Portuguese social security tax in considered one of the highest in Europe. However, the private-expenditure-to-GDP ratio in Portugal is similar to the threshold of 3% observed in the case of other EU countries retained in our analysis.

In Spain, the health care system is also funded mainly through taxes, and is dominated by the public sector [32]. As in Portugal, the Spanish health care system has recorded ample reforms over the past three decades, with a focus on the universal access to health, thus privileging the principles of the Beveridge model. However, unlike the Portuguese system, in Spain health competencies were transferred from the National Institute of Health to the regional level in 2002. In terms of financing, most of the funds come from public sector sources (70%), although the share of private sources considerably increased during the 1990s. Compared to the EU-15 members, Spain’s overall health care expenditure as a percentage of GDP is slightly below the average.

## 4. Methodology

This section presents the bound unit root tests and the tests for strict stationarity.

### 4.1. Bound Unit Root Tests

The development of the unit root tests for near-integrated time series proposed by [5,6] relies on a bounded unit root distribution, and starts from a discussion of asymptotic properties of some commonly used unit root tests (e.g., augmented Dickey–Fuller (ADF); Phillips–Perron (PP); Ng–Perron (MZ)). The authors propose new approaches for computing the critical values for these tests if the time series are bounded.

For example, Ref. [5] develops a PP unit root test for bounded series. If $X_{t}$ represents a stochastic process bounded between $\underline{b}$ and $\bar{b}$ (i.e., $X_{t}\in \left[\underline{b},\bar{b}\right]$, ∀ t), for each t the increment $∆X_{t}\in \left[\underline{b}-X_{t-1},\bar{b}-X_{t-1}\right]$. Ref. [6] builds upon [5], and considers a constant deterministic component in the analysis of bounded series; that is:(1)$$X_{t}=\theta +Y_{t}$$
and
(2)$$Y_{t}=\alpha Y_{t-1}+u_{t},\;\alpha =1$$
where (1) $u_{t}=∆X$ is decomposed ($u_{t}=\varepsilon_{t}+{\underline{\vartheta }}_{t}-{\bar{\vartheta }}_{t}$) and $\varepsilon_{t}$ follows an AR(1) process, (2) $\left\{{\underline{\vartheta }}_{t}\right\},\left\{{\bar{\vartheta }}_{t}\right\}$ are non-negative processes, and (3) the regulators ${\underline{\vartheta }}_{t}>0$ if $Y_{t-1}+\varepsilon_{t}>\underline{b}-\theta$, while ${\bar{\vartheta }}_{t}>0$ if $Y_{t-1}+\varepsilon_{t}>\bar{b}-\theta$.

The authors define $\underline{c}$ and $\bar{c}$ as measures impacting the bounds of finite-sample series, such as $\underline{b}=\underline{c}\lambda T^{1/2}$, $\bar{b}=\bar{c}\lambda T^{1/2}$, and $X_{0}=c_{0}\lambda T^{1/2}$, with $\underline{c}\leq c_{0}\leq \bar{c}$. The series’ first-order autoregressive coefficient is represented by ${\hat{\rho }}_{T}$, such that ${\hat{\rho }}_{T}\sum X_{t-1}^{2}=\sum X_{t-1}∆X_{t}$ and $t\in \left[1,T\right]$.

In the absence of bounded/integrated processes’ limits ($\underline{b}$ or $\bar{b}$)—that is, $\underline{c}=-\infty$ and $\bar{c}=+\infty$—${\hat{\rho }}_{T}$ has the asymptotic distribution $T\left({\hat{\rho }}_{T}-1\right)\overset{\omega }{\to }\frac{B{\left(1\right)}^{2}-\sigma^{2}/\lambda^{2}}{2\int_{0}^{1}B{\left(s\right)}^{2}ds}$ (with $\sigma^{2}:=\underset{T\to \infty }{\lim}T^{-1}\overset{T}{\underset{t=1}{\sum }}\varepsilon_{T}^{2}$). This asymptotic distribution becomes the unit root distribution $Z_{\rho }$ when $\lambda^{2}=\sigma^{2}$. Considering ${\hat{\lambda }}^{2},{\hat{\sigma }}^{2}$ as consistent estimators of $\lambda^{2},\sigma^{2}$, the PP-type test is:(3)$${\hat{Z}}_{\rho }:=T\left({\hat{\rho }}_{T}-1\right)-\frac{\frac{1}{2}\left({\hat{\lambda }}^{2}-{\hat{\sigma }}^{2}\;\right)}{T^{-2}\sum_{t=1}^{T}X_{t-1}^{2}}$$

In addition to the PP-type test, Ref. [5] proposes a Phillips’ modified t-test ${\hat{Z}}_{t}$:(4)$${\hat{Z}}_{t}:=\left(\hat{\sigma }/\hat{\lambda }\right)Z_{t}-\left\{T\left({\hat{\sigma }}^{2}-{\hat{\lambda }}^{2}\right)/\left(2\hat{\lambda }\right)\right\}{\left(\sum X_{t-1}^{2}\right)}^{-1/2}$$

More recently, Ref. [6] advanced two robust approaches for bounded-series unit root tests based on the Said–Dickey–Fuller test (ADF-type) and MZ statistics. Likewise, for a finite sample $X_{t}$, the ADF statistics can be derived from an ordinary least squares (OLS) regression:(5)$${\hat{X}}_{t}=\alpha {\hat{X}}_{t-1}+\overset{k}{\underset{i=1}{\sum }}\alpha_{i}∆{\hat{X}}_{t-i}+\varepsilon_{t,k}$$

The two ADF statistics computed by [6] are $ADF_{\alpha }≔\frac{T\left(\hat{\alpha }-1\right)}{\hat{\alpha }\left(1\right)}$ and $ADF_{t}≔\frac{\hat{\alpha }-1}{s\left(\hat{\alpha }\right)}$, where ${\hat{\alpha }}_{i}$ is the OLS estimator of $\alpha_{i}$ and $s\left(\hat{\alpha }\right)$ represents the standard error of $\hat{\alpha }$. At the same time, the MZ statistics are defined as $MZ_{\alpha }≔\frac{T^{-1}{\hat{X}}_{T}^{2}-T^{-1}{\hat{X}}_{0}^{2}-s_{AR}^{2}\left(k\right)}{2T^{-2}\sum_{t=1}^{T}{\hat{X}}_{t-1}^{2}}$, $M\&B≔{\left(T^{-2}\overset{T}{\underset{t=1}{\sum }}{\hat{X}}_{t-1}^{2}/s_{AR}^{2}\left(k\right)\right)}^{1/2}$, and $MZ_{t}≔\;MZ_{\alpha }\times M\&B$, where $s_{AR}^{2}\left(k\right)$ is an autoregressive estimator of spectral density.

For unbounded $X_{t}$ series (i.e., $\underline{b}=-\infty ,\bar{b}=+\infty$), the asymptotic (null) distributions of the ADF and MZ statistics are $ADF_{\alpha },MZ_{\alpha }\overset{\omega }{\to }\frac{1}{2}\left(F_{B}{\left(1\right)}^{2}-F_{B}{\left(0\right)}^{2}-1\right)\times {\left(\int_{0}^{1}F_{B}{\left(s\right)}^{2}ds\right)}^{-1}=:\zeta_{1}$, $M\&B\overset{\omega }{\to }{\left(\int_{0}^{1}F_{B}{\left(s\right)}^{2}ds\right)}^{\frac{1}{2}}=:\zeta_{2}$, and $ADF_{t},MZ_{t}\overset{\omega }{\to }\zeta_{3}≔\zeta_{1}\zeta_{2}$, where $F_{B}≔B-\int_{0}^{1}B\left(r\right)dr$ and B is a standard Brownian motion. 

In the presence of known bounds ($\underline{b},\bar{b}$), Ref. [6] defines consistent nuisance parameters as follows:(6)$$\hat{\underline{c}}≔\frac{\underline{b}-X_{0}}{s_{AR}\left(k\right)T^{1/2}},\;\hat{\bar{c}}≔\frac{\bar{b}-X_{0}}{s_{AR}\left(k\right)T^{1/2}}$$

Afterwards, [6] performs simulation tests and builds a càdlàg process $\;B_{n}^{*}$, such as $B_{n}^{*}\overset{\omega }{\to }B_{\underline{c}}^{\bar{c}}$, where $B_{\underline{c}}^{\bar{c}}$ is a regulated Brownian motion. The asymptotic distribution of the $ADF_{\alpha }$ test is:(7)$$ADF_{\alpha }^{*}≔\frac{{\tilde{X}}_{n}^{*}{\left(1\right)}^{2}-{\tilde{X}}_{n}^{*}{\left(0\right)}^{2}-1}{2\int_{0}^{1}{\tilde{X}}_{n}^{*}{\left(s\right)}^{2}ds}$$
where the càdlàg process is ${\tilde{X}}_{n}^{*}\left(s\right)≔X_{n}^{*}\left(s\right)-\int_{0}^{1}X_{n}^{*}\left(u\right)du$ and $s\in \left[0,1\right].$ The critical values of the unit root tests we use for bounded series, as well as the *p*-values, are generated as described in [5,6].

### 4.2. Strict Stationarity Tests 

For the purpose of robustness, we used a series of powerful tests relying on the null hypothesis of stationarity presence, called strict stationary tests. These tests represent variations of the well-known [33] *KPSS* test, and are designed for first-order stationarity, as follows:(8)$$KPSS=\frac{1}{{\left(\hat{w}T\right)}^{2}}{\sum_{k=1}^{T}\left(\sum_{t=1}^{k}\left(y_{t}-{\bar{y}}_{t}\right)\right)}^{2}$$
where ${\bar{y}}_{t}$ represents the mean of the ${\left\{y_{t}\right\}}_{t=1}^{T}$ sample and ${\hat{w}}^{2}$ is a nonparametric estimator of the long-run variance. 

First, Ref. [7] developed a robust version of the *KPSS* test (called the *IKPSS* test), considering the following empirical process:(9)$$I_{T}(r):=\frac{1}{\overset{⌢}{\sigma }\sqrt{T}}\sum_{t=1}^{\left[T_{r}\right]}sign\left(y_{t}-m_{T}\right)$$
where $m_{T}$ represents the median of ${\left\{y_{t}\right\}}_{t=1}^{T}$ and ${\hat{\sigma }}^{2}$ is a nonparametric consistent estimator of the long-run variance.

A Cramér–von Mises metric $h\left(\cdot \right)$ is proposed to measure the fluctuation of the empirical process $I_{T}\left(r\right)$, while the *IKPSS* test statistic becomes:(10)$$IKPSS:=\frac{1}{{\left(\overset{⌢}{\sigma }T\right)}^{2}}{\sum_{k=1}^{T}\left(\sum_{t=1}^{k}sign\left(y_{t}-m_{T}\right)\right)}^{2}$$

Compared to the classical *KPSS* test, the *IKPSS* test has a similar limiting distribution under the null of stationarity. However, under the alternative unit root hypothesis, *IKPSS* outperforms the *KPSS* test in the presence of fat-tailed errors, while in the case of thin tails it has a relatively lower power. In this context, Ref. [8] proposed a new test for second-order (covariance) stationarity, considering a standardised bivariate empirical process:(11)$$Z_{T}(r):=\frac{1}{\sqrt{T}}{\hat{\Omega }}^{-1/2}\sum_{t=1}^{\left[T_{r}\right]}\left(\begin{array}{l} {\tilde{y}}_{t}\\ v_{T} \end{array}\right)$$
where ${\tilde{y}}_{t}≔y_{T}-\frac{1}{T}\overset{T}{\underset{j-1}{\sum }}y_{j}$ represents demeaned data, $v_{t}≔{\tilde{y}}_{t}^{2}-\sigma_{y}^{2}$, $\sigma_{y}^{2}≔\frac{1}{T}\overset{T}{\underset{t=1}{\sum }}{\tilde{y}}_{t}^{2}$, and ${\;\hat{\Omega }}^{-1/2}$ is the inverse of the Choleski decomposition of the nonparametric consistent estimator ${\;\hat{\Omega }\;}^{2}$ of the long-run variance.

The nonparametric consistent estimator ${\;\hat{\Omega }\;}^{2}$ of the long-run variance is therefore:(12)$$\Omega^{2}=\underset{T\to \infty }{\lim}Ε\left[\left(\frac{1}{\sqrt{T}}\sum_{t=1}^{T}\left(\begin{array}{l} {\tilde{y}}_{t}\\ v_{t} \end{array}\right)\right){\left(\frac{1}{\sqrt{T}}\sum_{t=1}^{T}\left(\begin{array}{l} {\tilde{y}}_{t}\\ v_{t} \end{array}\right)\right)}^{’}\right]$$

The Kolmogorov metric is applied to measure the fluctuation of the empirical processes, and their test statistic (*XL* test) is defined as follows: (13)$$XL=\underset{1\leq k\leq T}{\max}{\left\|\frac{1}{\sqrt{T}}{\hat{\Omega }}^{-1/2}\sum_{t=1}^{k}\left(\begin{array}{l} {\tilde{y}}_{t}\\ v_{t} \end{array}\right)\right\|}_{1}$$

Unlike the *IKPSS* test, the *XL* test has a good power against alternative hypotheses of distribution varying in both time and distribution scale. Nevertheless, the power of the *XL* test is reduced when the distribution of a random variable varies in time, or when the fat tails are present. Therefore, in order to overcome this issue, Ref. [9] advances the hypothesis of strict stationarity by generalising the *IKPSS* test, using the sample quantiles and not the sample median. In this way, their test—called the *LN* test—has power against the unit root alternative, and good power in detecting changes in higher moments of the unconditional distribution.

Ref. [9] defines an empirical process as follows:(14)$$S_{T}(r,\tau ):=\frac{1}{\hat{\pi }(\tau )\sqrt{T}}\sum_{t=1}^{\left[Tr\right]}\psi_{\tau }(y_{t}-b(\tau ))$$
where $b\left(\tau \right)$ is the τth sample unconditional quantile of ${\left\{y_{t}\right\}}_{t=1}^{T}$, $r\in \left[0,1\right]$, and $\hat{\pi }{\left(\tau \right)}^{2}$ is a nonparametric consistent estimator of $\pi {\left(\tau \right)}^{2}$.

The estimator $\pi {\left(\tau \right)}^{2}\;$is expressed as follows:(15)$$\pi {\left(\tau \right)}^{2}:=\underset{T\to \infty }{\lim}E\left[{\left(\frac{1}{\sqrt{T}}\sum_{t=1}^{T}\psi_{\tau }(y_{t}-b_{0}(\tau ))\right)}^{2}\right]$$

The fluctuations of $S_{T}\left(r,\tau \right)$ across various quantiles $\tau \in \Gamma_{w}=\left[w,\;1-w\right]$ with $w\in \left(0,1/2\right)$ are measured using a Kolmogorov–Smirnoff metric. Thus, the *LN* test statistic for strict stationarity is defined as follows:(16)$$LN=\underset{\tau \in \Gamma_{w}}{\max}\underset{1\leq k\leq T}{\max}\frac{1}{\hat{\pi }(\tau )\sqrt{T}}\left|\sum_{t=1}^{k}\psi_{\tau }(y_{t}-b(\tau ))-\frac{k}{T}\sum_{t=1}^{T}\psi_{\tau }(y_{t}-b(\tau ))\right|$$
where $\hat{\pi }{\left(\tau \right)}^{2}$ is an HAC estimator.

The *K* kernel function of the HAC estimator is: (17)$$\hat{\pi }{\left(\tau \right)}^{2}:=\frac{1}{T}\sum_{i=1}^{T}\sum_{j=1}^{T}K\left(\frac{i-j}{q_{T}}\right)\psi_{\tau }(y_{i}-b(\tau ))\psi_{\tau }(y_{j}-b(\tau )),$$

## 5. Data and Results

### 5.1. Data

We used the annual data (1972–2019) for six EU countries, resorting to the OECD database (for testing the convergence, we relied on the spread from the group average, testing its stationarity). The sample selection was based on the availability of data. Our purpose was to provide a convergence analysis in terms of health car expenditure per capita and as percentage of GDP, relying on the longest data series possible. The OECD data regarding health expenditure are available starting from 1960, but this is not the case for the EU series, where the data are available starting from 1970 only for Austria, Finland, Germany, Portugal and Spain. Data for the Netherlands are available starting from 1971, while for other EU countries we can find data starting from 1975. Consequently, we decided to retain six countries for our analysis, and took 1972 as the starting year (the series stops in 2019). The six countries have different socioeconomic systems in place—namely, the Continental system (Austria and Germany), the Nordic system (Finland and the Netherlands), and the Southern system (Portugal and Spain)—but also different health care arrangements, as detailed in Section 3. 

The summary statistics are presented in Table 1.

The statistics show that Austria, Germany, and the Netherlands are placed above the group average, in terms of both overall expenditure per capita and as a percentage of GDP, while Finland, Portugal, and Spain are below the group average. This evidence persists for the governmental and private expenditure when we look to the health expenditure per capita. However, if we analyse the private expenditure as a percentage of GDP, Portugal is leading. We can also note that Finland and Portugal recorded the highest volatility levels in terms of health-care-expenditure-to-GDP ratio, while Germany showed the opposite trend.

### 5.2. Main Findings

In this section, we present two sets of results, considering health expenditure per capita and health expenditure as a percentage of GDP. For each set, we compare the convergence process by analysing the overall, public, and private expenditure (we use GLS demeaned series). Table 2 presents the findings of bounded unit root tests for overall health expenditure per capita.

We can see from Table 2 that convergence toward the group average is recorded in the case of Finland and Portugal when the overall and governmental health expenditure per capita are analysed. When we refer to the private expenditure per capita, convergence is noted only for Austria. The results are robust if we compare different bounded unit root tests. These results confirm the mixed findings previously reported in the literature (see, for example, [1,2,3]).

In Table 3, we report the convergence process results with respect to the health-care-expenditure-to-GDP ratio. Whether we consider the public, the private, or the overall expenditure as a percentage of GDP, we notice a convergence process only for Austria. These findings confirm the previous ones in the case of private expenditure, but not for the governmental and overall expenditure. In the case of governmental and private health expenditure as percentages of GDP, we report similar findings to those reported by [4] for an earlier period. 

In general, we can observe that the convergence process is weak and the significance (when it is the case) is only reached at a 90% confidence level. Therefore, our findings highlight the heterogeneity of the EU health care systems. Given the mixed findings we obtained, several robustness checks are necessary. To this end, we used a series of new proposed strict stationarity tests, as an alternative to bounded unit root tests. In next section, we apply these tests for both health expenditure per capita and as a percentage of GDP. 

### 5.3. Robustness Analysis Based on New Proposed Strict Stationarity Tests

When we refer to the health expenditure per capita, the results of the strict stationarity tests indicate the convergence process only for Austria, the Netherlands, and Finland. In the case of Finland, the convergence is achieved only in the case of overall expenditure per capita (Table 4). It should be noted that the null of these tests is the presence of stationarity (see the notes of Table 4 and Table 5). Consequently, these results confirm the main findings for Austria and Finland, but contradict the main findings in the case of the Netherlands (this result is hard to explain, especially for the private expenditure, given the implications of the 2006 reform implemented in this country). In addition, the tests for strict stationarity also indicate a convergence process in the case of Germany, for private expenditure per capita. 

Table 5 presents the results for the health expenditure as a % of GDP. We obtained similar findings to those reported in Table 4 with respect to the health expenditure per capita. The convergence process was recorded for Austria, Finland, and the Netherlands.

We can conclude, then, that the convergence process in terms of health expenditure as a percentage of GDP is limited in the selected EU countries. These results are in contrast with several findings reported in the literature (see, for example, [17]), but confirm the mixed evidence reported by recent papers (i.e., [4]). Our results can be explained by the heterogeneity of the EU health care systems. 

## 6. Conclusions and Policy Implications

After several reforms implemented in their health care systems, and many common initiatives designed to enhance the quality of medical care and to provide universal coverage, the EU countries still present strong heterogeneity in terms of health care expenditure. Therefore, only a portion of the previous research in this area has focused on testing the convergence process, whereas most of these works used stationarity tests to this end.

With a focus on a group of six EU countries, we add to this strand of the literature and make several contributions to the existing knowledge. First, resorting to bounded unit root tests, we used the group average as a benchmark to assess the health expenditure convergence process. Most previous works have considered the United States (US) as a benchmark for studying the convergence process in the EU. Even if the US leads the EU countries in terms of both health expenditure as a percentage of GDP and health expenditure per capita, using this benchmark is not recommended, given that the convergence process should happen within the EU countries. We posit that it is more useful to use the group average to investigate the trends and the effects of health care reforms within the EU. Second, in line with most previous papers, we checked the convergence process, considering the level of expenditure per capita. However, this indicator is highly influenced by the development level, and does not underline the public or the private budgetary effort necessary to reach a high quality and a convergence process. In this context, we compared the results of two sets of analyses, considering both health expenditure per capita and health expenditure as a percentage of GDP. The bounded unit root tests are particularly appealing in this case, given that the health expenditure as a percentage of GDP represents a bounded series—at least for the public sector. Lastly, we contributed to the existing literature by using a series of tests for strict stationarity for robustness purposes. As far as we know, this is the first paper to analyse the EU health expenditure convergence process using this category of tests.

In general, we obtained mixed findings regarding the convergence process. On the one hand, when we consider the health expenditure per capita, a convergence process is documented in the cases of Finland and Portugal, but only when the overall and public expenditure is considered. In terms of private expenditure, the convergence process is observed only for Austria. For all other countries included in our sample, we documented a non-stationary process, indicating a lack of convergence. On the other hand, when we assess the convergence in terms of the health-expenditure-to-GDP ratio, the convergence process is recorded for Austria only. The robustness check we performed using strict stationarity tests partially confirms the mixed results we obtained. The mixed evidence in terms of health expenditure convergence processes can be explained by the diversity and particularities of each national health care system. At the same time, although common efforts have been made to ensure universal coverage, individual EU countries have implemented their own strategies and financing procedures.

These results have three main policy implications: First, the heterogeneity of the EU health care systems requires the identification of common solutions to the EU health systems’ problems, in order to enhance their convergence process. These solutions involve effective strategies and efforts toward an integrated health education and research system. Improving the access to health data available to medical professionals is equally important. Second, a greater public financial effort is necessary to achieve the convergence process. For this purpose, the national authorities should ensure higher fiscal discipline, avoiding high budgetary deficits that impend the achievement of a convergence process. Third, the technology companies active in the health care sector should be encouraged to work together and to find solutions to create new products designed to improve quality of life.

Our paper has several limitations, represented by the reduced sample we used and by our mixed findings. Therefore, the empirical analysis can be extended as follows: On the one hand, several benchmarks can be used to test the convergence process. On the other hand, the sample can be extended, thus focusing on the recent period. Finally, a deeper analysis can be made using the club convergence process. The EU countries can be grouped depending on the characteristics of their health care systems, in order to investigate the convergence process within each group.

## Figures and Tables

**Figure 1 ijerph-19-01991-f001:**
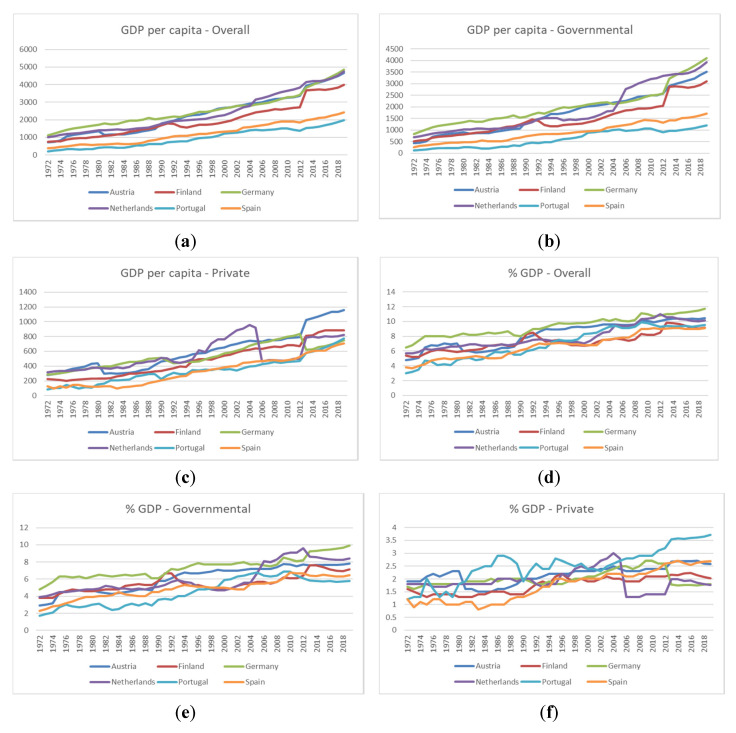
Health-expenditure-to-GDP ratio and health expenditure as % in GDP in selected countries. Source: OECD statistics.

**Table 1 ijerph-19-01991-t001:** Summary statistics.

Convergence ^1^	Austria	Finland	Germany	Netherlands	Portugal	Spain
Overall expenditure per capita						
MIN	12.55	−133.80	420.95	241.12	−1416.35	−970.65
MAX	685.50	355.64	789.52	1091.85	−487.75	−306.85
MEAN	366.51	10.53	611.65	445.05	−832.12	−587.76
SD	240.24	105.93	91.21	261.95	225.53	160.62
Governmental expenditure per capita						
MIN	−59.50	−155.78	317.47	33.97	−1240.07	−855.67
MAX	578.38	269.68	684.78	1235.93	−354.92	−212.70
MEAN	263.23	9.84	515.74	355.69	−699.48	−445.01
SD	206.64	114.35	101.27	359.49	209.50	160.54
Private expenditure per capita						
MIN	6.22	−52.00	−8.90	−144.15	−246.07	−189.27
MAX	177.35	68.35	200.57	316.23	−57.02	−91.83
MEAN	100.52	−2.08	95.92	86.59	−135.43	−145.52
SD	50.34	35.00	60.47	123.42	51.72	28.36
Overall expenditure as a % of GDP						
MIN	−0.40	−1.55	1.02	−0.97	−1.97	−1.68
MAX	1.37	0.93	2.22	1.32	0.52	−0.55
MEAN	0.49	−0.48	1.64	0.20	−0.77	−1.08
SD	0.53	0.73	0.37	0.55	0.76	0.24
Governmental expenditure as a % of GDP						
MIN	−0.43	−1.38	0.75	−1.15	−2.10	−1.42
MAX	1.37	1.21	2.12	2.22	0.27	−0.08
MEAN	0.38	−0.23	1.55	0.27	−1.16	−0.81
SD	0.48	0.73	0.42	0.85	0.73	0.29
Private expenditure as a % of GDP						
MIN	−0.23	−0.45	−0.38	−0.98	−0.37	−0.83
MAX	0.70	0.03	0.42	0.55	1.08	0.22
MEAN	0.14	−0.24	0.10	−0.05	0.41	−0.35
SD	0.23	0.12	0.20	0.45	0.39	0.28

^1^ The ratios of expenditure per capita and expenditure as a % of GDP are computed as a spread from the group average.

**Table 2 ijerph-19-01991-t002:** Results for bounded unit root tests—health expenditure per capita.

Convergence ^1^	Austria	Finland	Germany	Netherlands	Portugal	Spain
Overall expenditure per capita						
${\hat{Z}}_{\rho }$	−3.245	−6.431	−0.208	−1.787	−1.076	−2.473
${\hat{Z}}_{t}$	−1.305	−1.796 *	−0.164	−0.987	−0.801 **	−1.111
$ADF_{\alpha }^{*}$	−3.156	−8.954 *	−0.879	−1.498	−1.150 *	−3.229
$ADF_{t}^{*}$	−1.304	−2.092 *	−0.439	−0.923	−0.833 **	−1.310
$MZ_{\alpha }^{*}$	−3.035	−9.058 *	−1.124	−1.471	−1.134	−3.027
$MZ_{t}^{*}$	−1.254	−2.123 **	−0.609	−0.907	−0.822 **	−1.228
$M\&B^{*}$	0.413	0.234 *	0.542	0.616	0.725	0.406
Governmental expenditure per capita						
${\hat{Z}}_{\rho }$	−1.927	−5.562	−0.275	−3.092	−1.282 **	−5.267
${\hat{Z}}_{t}$	−1.041	−1.628	−0.183	−1.146	−0.826 ***	−1.729 *
$ADF_{\alpha }^{*}$	−1.432	−7.506 *	−1.218	−3.402	−1.090 **	−0.785
$ADF_{t}^{*}$	−0.791	−1.868 *	−0.622	−1.173	−0.776 ***	−0.544
$MZ_{\alpha }^{*}$	−1.860	−7.789 *	−0.994	−3.702	−1.076 **	−1.434
$MZ_{t}^{*}$	−1.025	−1.938 *	−0.516	−1.273	−0.765 ***	−1.024
$M\&B^{*}$	0.551	0.249 *	0.519	0.344	0.712	0.714
Private expenditure per capita						
${\hat{Z}}_{\rho }$	−8.727 **	−4.329	−2.557	−2.217	−3.035	−1.986
${\hat{Z}}_{t}$	−2.142 **	−1.416	−1.125	−0.713	−1.267	−0.856
$ADF_{\alpha }^{*}$	−7.161 *	−3.566	−2.263	−2.320	−3.685	−1.322
$ADF_{t}^{*}$	−1.976 **	−1.291	−1.071	−0.747	−1.406	−0.680
$MZ_{\alpha }^{*}$	−6.536 *	−3.411	−2.200	−2.254	−3.519	−0.616
$MZ_{t}^{*}$	−1.803 *	−1.234	−1.041	−0.726	−1.343	−0.376
$M\&B^{*}$	0.276	0.362	0.473	0.322	0.381	0.610

^1^ Notes: (1) the null hypothesis for all tests is the presence of the unit root; (2) ***, **, and * denote significance at 99%, 95%, and 90% confidence levels, respectively (series are stationary).

**Table 3 ijerph-19-01991-t003:** Results for bounded unit root tests – health expenditure as a % of GDP.

Convergence ^1^	Austria	Finland	Germany	Netherlands	Portugal	Spain
Overall expenditure as a % of GDP						
${\hat{Z}}_{\rho }$	−6.217 *	−1.798	−1.188	−2.539	−1.496	−4.023
${\hat{Z}}_{t}$	−1.822 *	−0.912	−0.629	−1.275	−0.974	−1.275
$ADF_{\alpha }^{*}$	−6.204	−1.179	−1.116	−1.781	−1.417	−4.728
$ADF_{t}^{*}$	−1.673 *	−0.549	−0.584	−1.024	−0.924	−1.318
$MZ_{\alpha }^{*}$	−4.988	−1.722	−1.132	−1.764	−1.468	−4.612
$MZ_{t}^{*}$	−1.693 *	−0.827	−0.542	−1.208	−0.907	−1.281
$M\&B^{*}$	0.371	0.531	0.588	0.552	0.483	0.328
Governmental expenditure as a % of GDP						
${\hat{Z}}_{\rho }$	−3.252	−1.492	−0.685	−3.234	−1.481	−4.255
${\hat{Z}}_{t}$	−1.562 *	−0.748	−0.366	−1.211	−0.824	−1.732 *
$ADF_{\alpha }^{*}$	−3.542	−0.892	−0.435	−1.946	−1.456	−0.196
$ADF_{t}^{*}$	−1.536 *	−0.467	−0.256	−0.926	−0.825	−0.232
$MZ_{\alpha }^{*}$	−3.286	−1.512	−0.456	−1.857	−1.423	−0.598
$MZ_{t}^{*}$	−1.571 *	−0.865	−0.267	−0.883	−0.643	−0.725
$M\&B^{*}$	0.425	0.448	0.594	0.547	0.582	0.870
Governmental expenditure as a % of GDP						
${\hat{Z}}_{\rho }$	−5.706 *	−7.668 *	−5.112	−2.415	−2.978	−2.622
${\hat{Z}}_{t}$	−1.728 *	−2.158 **	−1.623 *	−0.772	−1.364	−0.916
$ADF_{\alpha }^{*}$	−4.725	−2.989	−5.629 *	−2.469	−4.715	−3.612
$ADF_{t}^{*}$	−1.668 *	−1.420	−1.630	−0.616	−1.565 *	−1.124
$MZ_{\alpha }^{*}$	−4.327	−3.424	−5.245	−2.304	−4.425	−3.520
$MZ_{t}^{*}$	−1.475	−1.264	−1.724 *	−0.736	−1.499	−1.132
$M\&B^{*}$	0.380	0.395	0.322	0.345	0.473	0.321

^1^ Notes: (1) the null hypothesis for all tests is the presence of the unit root; (2) ***, **, and * denote significance at 99%, 95%, and 90% confidence levels, respectively (series are stationary).

**Table 4 ijerph-19-01991-t004:** Results of the tests for strict stationarity—health expenditure per capita.

Convergence ^1^	Austria	Finland	Germany	Netherlands	Portugal	Spain
Overall expenditure per capita						
*KPSS*	0.531	0.234	1.053 ***	0.289	1.016 ***	0.961 ***
*IKPSS*	0.440	0.102	0.991 **	0.273	0.974 **	0.899 *
*XL*	1.606	1.331	2.128 **	1.141	1.923 *	2.003 **
*LN*	1.479 *	1.243	1.691 **	1.114	1.696 ***	1.697 ***
Governmental expenditure per capita						
*KPSS*	0.719 *	0.426	1.042 ***	0.235	0.940 **	0.789 **
*IKPSS*	0.730 *	0.745 **	0.921 **	0.265	0.985 ***	0.746 **
*XL*	1.567	2.015 **	2.202 **	1.806	2.065 **	1.685
*LN*	1.682 ***	1.659 ***	1.670 **	1.391	1.691 ***	1.567 **
Private expenditure per capita						
*KPSS*	0.147	0.822 **	0.446	0.582 *	0.513 *	0.910 **
*IKPSS*	0.163	0.945 ***	0.430	0.306	0.188	0.961 **
*XL*	1.695	1.835	1.653	1.928	1.859 **	1.904
*LN*	1.273	1.554*	1.415	1.391	1.455 **	1.697 ***

^1^ Notes: (1) the null hypothesis for all tests is the presence of stationarity; (2) ***, **, and * denote significance at 99%, 95%, and 90% confidence levels, respectively (series are non-stationary).

**Table 5 ijerph-19-01991-t005:** Results of the tests for strict stationarity—health expenditure as a % of GDP.

Convergence ^1^	Austria	Finland	Germany	Netherlands	Portugal	Spain
Overall expenditure as a % of GDP						
*KPSS*	0.200	0.851 **	0.981 ***	0.352	1.024 ***	0.858 ***
*IKPSS*	0.128	0.911 **	0.762 **	0.228	0.950 **	0.671 *
*XL*	1.548	1.971 *	1.864	1.366	1.925	1.551
*LN*	1.363	1.616 **	1.670 **	1.177	1.682 **	1.646 ***
Governmental expenditure as a % of GDP						
*KPSS*	0.543	0.864 **	0.936 ***	0.203	0.947 **	0.295
*IKPSS*	0.478	0.920 **	0.795 **	0.258	1.054 ***	0.128
*XL*	1.596	1.922 *	1.832	1.753	1.963 **	1.662 *
*LN*	1.373	1.696 ***	1.657 **	1.348	1.691 ***	1.255
Private expenditure as a % of GDP						
*KPSS*	0.305	0.140	0.249	0.611 **	0.282	0.917 **
*IKPSS*	0.242	0.238	0.173	0.072	0.260	0.961 **
*XL*	1.839 *	1.764 *	1.196	1.892	1.934 **	1.890
*LN*	1.439 **	1.271	1.182	1.416 *	1.422 *	1.697 ***

^1^ Notes: (1) The null hypothesis for all tests is the presence of stationarity; (2) ***, **, and * denote significance at 99%, 95%, and 90% confidence levels, respectively (series are non-stationary).

## Data Availability

Data were extracted from the OECD database, and can be provided by the author upon request.

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
