# Peer review of "Health Care Expenditure in the European Union Countries: New Insights about the Convergence Process"

_ijerph, 2022, doi:10.3390/ijerph19041991_

Round 1
Reviewer 1 Report
The manuscript “Health care expenditures in the EU countries: new insights 2
about the convergence process ” used an advanced econometric models that show the convergence in the health expenditures for selected countries . Apparently the author use robust methods of analysis. However, my main comment is that the author should ponder some aspects for example methodological aspects vs a clear message in public health implications.
I have some other comments enumerated below.
Title: EU is not an international abbreviation, it can be substituted by “european” countries
Tables -It is missing the unit account of currency
Table 1- the Word “convergence” it is not informative
Conclusions and policy implications
I suggest the author should review the conclusion section; particularly limitations and policy implication are poor.
Author Response
Response to reviewer’s comments
Original title: Health care expenditures in the EU countries: new insights about the convergence process
New proposed title following the referee comments: Health care expenditures in the European Union countries: new insights about the convergence process
Reviewer #1
Thank you for providing me the opportunity to revise and resubmit my paper for publication in International Journal of Environmental Research and Public Health. I have modified the manuscript according to the reviewers’ report. The way I address each comment is detailed below, point-by-point. I am aware that the revision improved the quality of the paper and I thanks the reviewers for their effort to evaluate the manuscript. I have marked in blue color all the modifications made to the original text.
The manuscript “Health care expenditures in the EU countries: new insights 2
about the convergence process ” used an advanced econometric models that show the convergence in the health expenditures for selected countries. Apparently the author use robust methods of analysis. However, my main comment is that the author should ponder some aspects for example methodological aspects vs a clear message in public health implications.
Author’s response: Thank you for this comment. Indeed, to target a broader audience, the paper should have a clear message in terms of public health implications. In the second paragraph of the Introduction, it is mentioned that “The issue of health expenditures convergence in the EU is therefore doubly important. Firstly, the analysis of the convergence of health care expenditures allows assessing the outcome of common policies implemented in this area and particularly the level of the harmonization in terms of services quality provided by different health care systems. Secondly, a thorough analysis of the convergence process contributes to identify potential risk triggered by a migration to high quality health care services inside the EU.”
In the revised version of the paper I have added some explanation to underline why the study of convergence process is important and which are the implications of the existence/absence of the convergence process:
” Consequently, the documentation of a convergence process in the EU health care systems provides noteworthy information about the implementation and effectiveness of EU health care policies by the EU member states. In addition, the convergence in health care –a mainstream approach to healthcare research– means a harmonization in terms of edu-cation, medical research and capacity to mitigate the risks, as those triggered by the recent sanitary crisis. The absence of convergence simply show that human health issues are differently addressed among EU countries, and the fundamental human rights, namely the access of people to the health services they need, are not completely achieved. For ex-ample, in the context of COVID-19 pandemics, we have witnessed a different pressure on the national healthcare systems, although the health outcomes in the EU improved during the last decade. Different capacities in terms of intensive care units and very different mortality rates rise questions about the intensification of the convergence process inside the EU. “
Thank you for this comment!
I have some other comments enumerated below.
Title: EU is not an international abbreviation, it can be substituted by “european” countries
Author’s response: Done, thank you!
Tables -It is missing the unit account of currency
Author’s response: We have worked with ratios (e.g. expenditures/GDP; expenditures/capita) directly extracted from OECD database. Therefore, in Table 1 we have the summary statistics for the spread of those ratios from the group average. In the other tables, we have coefficients of tests and not currency units.
Table 1- the Word “convergence” it is not informative
Author’s response: I have adjusted the note of Table 1 to provide information about how the convergence process is tested. Thank you!
Conclusions and policy implications
I suggest the author should review the conclusion section; particularly limitations and policy implication are poor.
Author’s response: Thank you for this comment! Identifying specific policy recommendations is always challenging for the researchers. I have done my best to extend the policy implications starting from the mixed results I have obtained, rather pointing to a lack of convergence process. This paragraph was extended as follows:
“These results have three main policy implications. First, the heterogeneity of the EU health care systems requires the identification of common solutions to the EU health systems’ problems in order to enhance their convergence process. These solutions involves effective strategies and efforts toward an integrated health education and research system. Improving the access to health data available to medical professionals is equally important. Second, a higher public financial effort is necessary to achieve the convergence process. For this purpose, the national authorities should ensure a higher fiscal discipline, avoiding high budgetary deficits that impend the achievement of a convergence process. Third, the technology companies active in the health care sector should be encouraged to work together and to find solutions to create new products designed to improve the quality of life.”
In addition, I have mentioned in what consists the limitations of the paper:
“Our paper has several limitations, represented by the reduced sample we use and by the mixed findings we report. Therefore, the empirical analysis can be extended as follows”.
Thank you for all comments and suggestions!
Reviewer 2 Report
General remark: This is a technical article covering sophisticated mathematics. The quality of the paper is very good.
The description of countries as UE countries is misleading – not all did belong to EU since its beginning. Maybe it is worth to mention when they become a part of EU (join it)
72: “characterizes de heterogeneity” – not clear
190: “(main reforms are presented in Appendix)” – there is no Appendix, they are described earlier, in the text
The only weaker part (in my opinion) is a description of motivation – what Authors wanted to achieve. In my opinion a comparison is not enough. Please convince the reader that your research is necessary and has an utilitarian value.
Author Response
Response to reviewer’s comments
Original title: Health care expenditures in the EU countries: new insights about the convergence process
New proposed title following the referee comments: Health care expenditures in the European Union countries: new insights about the convergence process
Reviewer #2
Thank you for providing me the opportunity to revise and resubmit my paper for publication in International Journal of Environmental Research and Public Health. I have modified the manuscript according to the reviewers’ report. The way I address each comment is detailed below, point-by-point. I am aware that the revision improved the quality of the paper and I thanks the reviewers for their effort to evaluate the manuscript. I have marked in blue color all the modifications made to the original text.
General remark: This is a technical article covering sophisticated mathematics. The quality of the paper is very good.
Author’s response: Thank you for having appreciated my paper!
The description of countries as UE countries is misleading – not all did belong to EU since its beginning. Maybe it is worth to mention when they become a part of EU (join it)
Author’s response: Thank you, I have now mentioned in the footnote 1 that: “Although all these countries represent “old” EU member states, their accession to the EU happened at different moments in time. For example, in 1972 only Germany and Netherlands were part of the EU, whereas Portugal and Spain joined the Union in 1986. Finally, Austria and Finland joined the EU in 1995.”
72: “characterizes de heterogeneity” – not clear
Author’s response: I have rephrased this paragraph as follows: “Although the health care sectors of these countries rely on the “Bismarckian” tradition, they are very heterogeneous (for more details, please refer to Section 3).”
Thank you!
190: “(main reforms are presented in Appendix)” – there is no Appendix, they are described earlier, in the text
Author’s response: Thank you very much for signalizing this omission!
I have now inserted in Appendix a table that contains the main reforms undergone by the health care sectors of the six EU countries.
The only weaker part (in my opinion) is a description of motivation – what Authors wanted to achieve. In my opinion a comparison is not enough. Please convince the reader that your research is necessary and has an utilitarian value.
Author’s response: I have tried to underline in Introduction why the investigation of the health care sector convergence process warrants to be investigated, especially in the case of EU countries. In the revised version of the paper, to be more convincing, I have extended this part (the second paragraph of Introduction) as follows:
“Consequently, the documentation of a convergence process in the EU health care systems provides noteworthy information about the implementation and effectiveness of EU health care policies by the EU member states. In addition, the convergence in health care –a mainstream approach to healthcare research– means a harmonization in terms of edu-cation, medical research and capacity to mitigate the risks, as those triggered by the recent sanitary crisis. The absence of convergence simply show that human health issues are differently addressed among EU countries, and the fundamental human rights, namely the access of people to the health services they need, are not completely achieved. For ex-ample, in the context of COVID-19 pandemics, we have witnessed a different pressure on the national healthcare systems, although the health outcomes in the EU improved during the last decade. Different capacities in terms of intensive care units and very different mortality rates rise questions about the intensification of the convergence process inside the EU.”
Thank you for all comments and suggestions!
Reviewer 3 Report
The paper is interesting and highlights an important topic, with significant policy links. Those links need highlighting, especially in relation to European integration and European health policies. The selection of case studies needs to be explained.
Author Response
Response to reviewer’s comments
Original title: Health care expenditures in the EU countries: new insights about the convergence process
New proposed title following the referee comments: Health care expenditures in the European Union countries: new insights about the convergence process
Reviewer #3
Thank you for providing me the opportunity to revise and resubmit my paper for publication in International Journal of Environmental Research and Public Health. I have modified the manuscript according to the reviewers’ report. The way I address each comment is detailed below, point-by-point. I am aware that the revision improved the quality of the paper and I thanks the reviewers for their effort to evaluate the manuscript. I have marked in blue color all the modifications made to the original text.
The paper is interesting and highlights an important topic, with significant policy links. Those links need highlighting, especially in relation to European integration and European health policies. The selection of case studies needs to be explained.
Author’s response: Thank you for your positive comment. In the revised version of the paper the importance of investigating the convergence process of EU health care sectors is better highlighted. The second paragraph becomes:
“The issue of health expenditures convergence in the EU is therefore doubly important. Firstly, the analysis of the convergence of health care expenditures allows assessing the outcome of common policies implemented in this area and particularly the level of the harmonization in terms of services quality provided by different health care systems. Secondly, a thorough analysis of the convergence process contributes to identify potential risk triggered by a migration to high quality health care services inside the EU, or by a global pandemic. Consequently, the documentation of a convergence process in the EU health care systems provides noteworthy information about the implementation and effectiveness of EU health care policies by the EU member states. In addition, the convergence in health care –a mainstream approach to healthcare research– means a harmonization in terms of education, medical research and capacity to mitigate the risks, as those triggered by the recent sanitary crisis. The absence of convergence simply show that human health issues are differently addressed among EU countries, and the fundamental human rights, namely the access of people to the health services they need, are not completely achieved. For example, in the context of COVID-19 pandemics, we have witnessed a different pressure on the national healthcare systems, although the health outcomes in the EU improved during the last decade. Different capacities in terms of intensive care units and very different mortality rates rise questions about the intensification of the convergence process inside the EU.”
Further, I have tried to motivate the selection of six EU countries, behind the data availability constraints:
“Third, we analyse the convergence process among the above-mentioned six EU countries for three reasons. Although the health care sectors of these countries rely on the “Bismarckian” tradition, they are very heterogeneous (for more details, please refer to Section 3). Moreover, we use Organisation for Economic Co-operation and Development (OECD) statistics, and the selection of countries is constrained by data availability. Our purpose is to study the convergence process on the longest available timespan and to include in the analysis the beginning of the 1970s when ample reforms were carried out in the EU countries in terms of health care systems (for additional information regarding the choice of data sample please refer to Section 3.1.).”
Thank you for all comments and suggestions!
Round 2
Reviewer 1 Report
Thank you for providing a revised version of the manuscript. In my opinion, all points have been addressed and the quality of the manuscript improved.